# Reinducing Radioiodine-Sensitivity in Radioiodine-Refractory Thyroid Cancer Using Lenvatinib (RESET): Study Protocol for a Single-Center, Open Label Phase II Trial

**DOI:** 10.3390/diagnostics12123154

**Published:** 2022-12-14

**Authors:** Maaike Dotinga, Dennis Vriens, Floris H. P. van Velden, Mette K. Stam, Jan W. T. Heemskerk, Petra Dibbets-Schneider, Martin Pool, Daphne D. D. Rietbergen, Lioe-Fee de Geus-Oei, Ellen Kapiteijn

**Affiliations:** 1Department of Radiology, Section of Nuclear Medicine, Leiden University Medical Center, 2333 ZA Leiden, The Netherlands; 2Department of Clinical Pharmacology and Toxicology, Leiden University Medical Center, 2333 ZA Leiden, The Netherlands; 3Biomedical Photonic Imaging Group, University of Twente, 7522 NB Enschede, The Netherlands; 4Department of Medical Oncology, Leiden University Medical Center, 2333 ZA Leiden, The Netherlands

**Keywords:** thyroid cancer, I-124 dosimetry, lenvatinib, radioiodine uptake, redifferentiation, radioiodine-refractory DTC

## Abstract

**Background**: Management of patients with radioiodine (RAI)-refractory differentiated thyroid cancer (DTC) is a challenge as I-131 therapy is deemed ineffective while standard-of-care systemic therapy with tyrosine kinase inhibitor (TKI) lenvatinib is associated with frequent toxicities leading to dose reductions and withdrawal. A potential new treatment approach is to use TKIs as redifferentiation agent to restore RAI uptake to an extent that I-131 therapy is warranted. Prior studies show that short-term treatment with other TKIs restores RAI uptake in 50–60% of radioiodine-refractory DTC patients, but this concept has not been investigated for lenvatinib. Furthermore, the optimal duration of treatment with TKIs for maximal redifferentiation has not been explored. **Methods and Design**: A total of 12 patients with RAI-refractory DTC with an indication for lenvatinib will undergo I-124 PET/CT to quantify RAI uptake. This process is repeated after 6 and 12 weeks post-initiating lenvatinib after which the prospective dose estimate to target lesions and organs at risk will be determined. Patients will subsequently stop lenvatinib and undergo I-131 treatment if it is deemed effective and safe by predefined norms. The I-124 PET/CT measurements after 6 and 12 weeks of the first six patients are compared and the optimal timepoint will be determined for the remaining patients. In all I-131 treated patients post-therapy SPECT/CT dosimetry verification will be performed. During follow-up, clinical response will be evaluated using serum thyroglobulin levels and F-18 FDG PET/CT imaging for 6 months. It is hypothesized that at least 40% of patients will show meaningful renewed RAI uptake after short-term lenvatinib treatment. **Discussion**: Shorter treatment duration of lenvatinib treatment is preferred because of frequent toxicity-related dose reductions and drug withdrawals in long-term lenvatinib treatment. Short-term treatment with lenvatinib with subsequent I-131 therapy poses a potential new management approach for these patients. Since treatment duration is reduced and I-131 therapy is more tolerable for most patients, this potentially leads to less toxicity and higher quality of life. Identifying RAI-refractory DTC patients who redifferentiate after lenvatinib therapy is therefore crucial. **Trial Registration**: ClinicalTrials.gov, NTC04858867.

## 1. Background

Thyroid cancer is the most common endocrine malignancy, accounting for approximately 3.1% of all cancer diagnoses worldwide and an estimated 41,000 deaths in 2018 [1]. The majority of cases are suffering from differentiated thyroid cancer (DTC), which is commonly managed by total thyroidectomy and subsequent radioiodine-131 (I-131) therapy and is associated with excellent long-term survival [2,3]. The effectiveness of I-131 therapy is dependent on the ability of cancer tissue to take up and retain radioiodine (RAI) after which the tissue structure is damaged by radiation. The sodium iodide symporter (NIS) plays a pivotal role in the uptake and retention of iodine in thyrocytes and thyroid cancer cells [4,5,6,7].

The function to take up and retain iodine is initially or gradually lost in 5–15% of all DTC cases and up to 50% of cases with metastases [6,8], indicating a state of dedifferentiation that is known as RAI-refractory disease [7]. To prevent futile repeated I-131 therapy and consider more effective treatments, it is crucial to identify RAI-refractory DTC early. Currently, diagnosis is merely based on imaging and response to I-131 therapy [9]. While criteria have been developed over time, the definition of RAI-refractory DTC is still controversial. The disease can be diagnosed when a patient complies to at least one of the following criteria: no RAI uptake present on diagnostic or therapeutic RAI scintigraphy (in some or all known lesions), progression of metastases despite RAI uptake or patients reaching a cumulative prescribed I-131 activity of >22.2 GBq [9,10]. In all RAI-refractory DTC patients, I-131 therapy is no longer deemed effective. One major pitfall in the current definition is the lack of assessment of the extent of RAI uptake by the lesions. This would require a dosimetry approach but is not yet embedded in routine clinical practice.

Management of RAI-refractory DTC patients is a considerable challenge in clinical practice. Therefore, the disease is associated with a poor overall prognosis: the 10-year survival rate is less than 10% and the mean life expectancy only 3–5 years [11]. Systemic therapy with the oral multi-receptor tyrosine kinase inhibitor (TKI) lenvatinib is approved for progressive RAI-refractory DTC and currently standard practice. Further systemic treatment options are limited. Though a response rate of 65% and improved median progression-free survival in comparison to placebo (18 months versus 3.6 months) is observed [12], the toxicity burden associated with lenvatinib should be taken into consideration, as this may significantly compromise the patients’ quality of life [4,13]. Treatment should, therefore, continue as long as clinical benefit is observed or until unacceptable toxicity occurs [10]. According to real-life data, toxicity is frequently observed leading to dose reductions in 56% of patients and toxicity-related drug withdrawal in 44% of patients [14].

A potential alternative way to achieve disease control in RAI-refractory thyroid cancer is to restore RAI uptake using TKIs as redifferentiation agents [15]. The re-induced RAI uptake enables the possibility to subsequently treat patients with I-131, thereby reducing the administration duration of TKIs and subsequent adverse events. Several clinical studies have already shown the ability of TKIs, including selumetinib, trametinib, vemurafenib, dabrafenib and larotrectinib to re-induce RAI uptake in RAI-refractory thyroid cancer patients [16,17,18,19,20]. These studies showed that after short-term therapy of approximately 4 weeks with these TKIs, renewed or enhanced RAI uptake was observed in 35–67% of RAI-refractory patients. Subsequently, I-131 therapy was initiated in 35–71% of all patients [16,17,18,19,20,21,22,23].

The TKIs for which this redifferentiation effect is observed are all selectively targeting either mitogen-activated protein kinase kinase (MEK), B-type rapidly accelerated fibrosarcoma kinase (BRAF) mutations or neurotrophic tyrosine receptor kinase rearrangements, thereby downstream inhibiting the mitogen-activated protein kinase (MAPK) pathway. In contrast, lenvatinib inhibits multiple receptor tyrosine kinases including vascular endothelial growth factor receptor (VEGFR1-3), fibroblast growth factor receptor (FGFR1-4), platelet-derived growth factor receptor α (PDGFRα) and rearranged during transfection (RET), further inhibiting induced signaling through the MAPK and PI3K/Akt pathways [24]. As aberrant activation of MAPK and phosphatidylinositol-3-kinase (PI3K)/protein kinase B (Akt) pathways inhibits NIS transcription and is associated with decreased NIS messenger ribonucleic acid (mRNA) expression [4], lenvatinib is hypothesized to restore functional NIS expression and thereby enhance iodine accumulation. An in vitro study by Anschlag et al. showed that technetium-99 m uptake (as a surrogate for NIS expression) was increased with a maximum of 326% in a well-differentiated human papillary thyroid carcinoma cell line after exposure to lenvatinib, but smaller effects were observed in poorly differentiated thyroid carcinoma cell lines [25]. They state that ‘the final proof for a redifferentiating potential of lenvatinib is incomplete’ based on their studies [25]. As experimental TKIs are currently not available to all patients, in contrast to the approved lenvatinib that is already part of standard clinical care and embedded in current guidelines [10], we believe evaluating its redifferentiation effect is very useful as it could prove to be a game-changer in the management of RAI-refractory DTC. Little is known about how long this redifferentiation effect lasts and whether it increases or decreases with increasing treatment duration. It is therefore of interest to investigate whether duration of treatment is of influence in the extent of renewed or increased RAI uptake after lenvatinib treatment in this trial.

The focus in this trial will be to evaluate the ability of lenvatinib to redifferentiating RAI-refractory DTC to an extent that functional RAI uptake is restored and I-131 therapy is justified. We will objectify this by quantitative molecular imaging and will predict efficacy and toxicity of radioiodine-131 based on dosimetry. We hypothesize that at least 40% of the patients will show sufficient redifferentiation for subsequent meaningful I-131 therapy. Since this would imply reduced lenvatinib treatment duration and since I-131 therapy is generally better tolerated by patients, this potentially leads to a new, less toxic treatment approach with higher quality-of-life.

## 2. Methods and Design

### 2.1. Study Design

This is a single-center open label phase II study evaluating the effect of short-term lenvatinib treatment for restoring RAI uptake and retention in RAI-refractory thyroid cancer to warrant I-131 therapy.

### 2.2. Trial Organization and Coordination

The RESET trial has been designed by the study initiators at the Department of Medical Oncology and the Department of Radiology, section Nuclear Medicine, at the Leiden University Medical Center (LUMC). Both departments are responsible for the overall trial management, database management and quality assurance including monitoring and reporting. The Department of Clinical Pharmacy and Toxicology is responsible for dispensing and quality assurance regarding the use of I-124.

### 2.3. Ethics, Informed Consent and Monitoring

This study has been approved by the Medical Ethics Committee Leiden-Den Haag-Delft (P20.096) and is registered on ClinicalTrials.gov (NTC04858867). This study complies with the Helsinki Declaration, the Medical Association code of conduct, the principles of Good Clinical Practice (GCP) and the General Data Protection Regulation. This trial will be carried out following local legal and regulatory requirements. All patients must declare their informed consent to participate in the trial in written form before enrollment. An independent monitoring committee appointed by the LUMC will perform on-site monitoring of recruited patients according to the International Conference on Harmonization (ICH) Harmonized Tripartite Guideline for GCP.

### 2.4. Sample Size Calculation

The study requires 12 patients of which 5 (42%) are expected to show meaningful renewed uptake of radioiodine to confirm that the real fraction is at least 15% (binomial distribution 95%-confidence interval: 15.2–72.3%).

### 2.5. Study Overview

Patients with RAI-refractory (determined by the multidisciplinary tumor board) DTC with the intention of starting standard-of-care lenvatinib treatment will be included in this study. Prior to lenvatinib treatment, patients will undergo recombinant human thyroid-stimulating hormone TSH (rhTSH)-stimulated I-124 dosimetry to quantify RAI uptake and retention at baseline. Besides that, patients undergo F-18 fluorodeoxyglucose (FDG) positron emission tomography (PET)/computed tomography (CT) and a biopsy is performed. An additional biopsy is performed after 6 weeks of lenvatinib treatment. The first half of the intended sample size (cohort 1, N = 6) will be treated with lenvatinib for a total of 12 weeks (Figure 1). After 6- and 12-week treatment, patients will undergo rhTSH-stimulated I-124 dosimetry to evaluate the redifferentiation effect and to assess expected absorbed lesion doses and the maximum tolerable activity (MTA). Patients will undergo subsequent rhTSH-stimulated I-131 therapy with the MTA (or a maximum of 7.4 GBq) if a clinically meaningful lesion dose (≥20 Gy) is expected. For all patients eligible for I-131 therapy, lenvatinib is discontinued prior to administration of I-131 and post-therapeutic I-131 dosimetry will be performed for dose verification. Results between 6- and 12-week lenvatinib treatment will be compared to select the lenvatinib treatment duration that leads to highest extent of redifferentiation. The next 6 patients (cohort 2) will then receive lenvatinib for either 6 or 12 weeks. Patients who are not eligible for I-131 therapy, will continue lenvatinib treatment at the discretion of the treating physician, effectively undergoing standard-of-care treatment. Patients will be followed up according to current guidelines for a total of 9 months after start of lenvatinib treatment. In case of progression after RAI, lenvatinib treatment will be restarted as standard of care. Metabolic and biochemical response will be assessed using F-18 FDG PET/CT and thyroglobulin (Tg) levels, respectively. An overview of all study procedures is given in Table 1.

### 2.6. Study Endpoints

The primary endpoint of the study is the fraction of RAI-refractory thyroid cancer patients that is eligible for I-131 therapy after 6- or 12-week lenvatinib treatment to an extent that clinically meaningful tumor radiation doses can be safely delivered with acceptable I-131 doses as determined by I-124 dosimetry of both lesions and healthy organs at risk.

Secondary endpoints include:Extent of RAI uptake at baseline and after 6- or 12-week lenvatinib;Optimal duration of lenvatinib treatment (6 weeks or 12 weeks) for maximum redifferentiation;Agreement between expected absorbed dose per lesion predicted by I-124 PET/CT dosimetry and actual absorbed dose per lesion determined by intra-therapeutic I-131 single photon emission computed tomography (SPECT)/CT dosimetry in patients in which I-131 therapy is warranted;Metabolic and biochemical treatment response using F-18 FDG PET and unstimulated (thyroid-stimulating hormone (TSH)-suppressed) Tg levels, respectively;Progression free survival, best objective response and overall survival;Incidence and severity of toxicities according to the Common Terminology Criteria for Adverse Events (CTCAE) 5.0;Quality of life (QoL).

An explorative endpoint of this study is to evaluate alterations at the transcriptional and translational level in biopsied tumor lesions before and after 6-week lenvatinib treatment and to determine whether treatment response is related to genetical profiles.

### 2.7. Patient Selection

Eligible patients will be screened by the treating medical oncologist. Prior to initiation of lenvatinib treatment, physical exam and blood sampling will be performed and vital signs, weight and performance status will be assessed to evaluate whether patients are eligible for the treatment and study. The first 12 patients conforming to inclusion and exclusion criteria (Table 2) agreeing to participate will be enrolled in the study.

### 2.8. Study Procedures

#### 2.8.1. Lenvatinib Treatment

Standard-of-care lenvatinib treatment will be provided to all patients included in this study for a duration of 6 or 12 weeks. A starting dose of 20 or 24 mg daily will be given in oral capsules. Dose reductions or interruptions will be induced at the treating physicians’ discretion in case of intolerable adverse events despite optimal management. In case of missed or interrupted administrations of lenvatinib, rescheduling of study procedures will be considered.

#### 2.8.2. Biopsy

A biopsy will be performed at baseline and an additional biopsy will be performed in the same lesion after 6 weeks of lenvatinib treatment. Lesions should be accessible for biopsy with low risk of severe side effects to the patients. Identification of the target lesions depends on the discretion of the treating physician. Biopsied tissue specimens will be evaluated with genomic, transcriptomic and protein-based research techniques to analyze whether alterations in molecular expression occur post-lenvatinib treatment and if treatment response is related to genetical profiles.

#### 2.8.3. I-124 Dosimetry Procedures

For cohort 1, rhTSH-stimulated I-124 dosimetry will be performed at baseline and after 6- and 12-week lenvatinib treatment. For cohort 2, rhTSH-stimulated I-124 dosimetry will be performed at baseline and at the most optimal interval observed in cohort 1 (either after 6- or 12-weeks lenvatinib treatment). Patients continue lenvatinib treatment during the procedures. Patients adhere to a low iodine diet 7 days prior to I-124 administration until 24 h post-administration. rhTSH injections will be given to the patient at home during 2 consecutive days (0.9 mg i.m.) prior to I-124 administration. Thyroid hormone (over)substitution may be continued. On the third day, but before administration of I-124, serum TSH and free thyroxine (fT4) will be measured as well as beta-human chorionic gonadotropin (HCG) in women of child-bearing potential. A total of 37 MBq (±10%) I-124 will be administered orally if TSH levels are >30 mU/L and beta-HCG is negative.

I-124 dosimetry procedures include:I-124 PET/CT
Pre-treatment dosimetry with PET/CT will be performed according to Jentzen et al. [26] This approach requires two PET/CT scans at 24 ± 6 and 96–120 h post-administration.
Activity in blood and the whole body
To assess toxicity of the bone marrow and lungs, the activity in blood samples and the whole body is determined according to Jentzen et al. [27] One heparinized blood sample of 2 mL is withdrawn from the patient and whole body counts measured at 2 ± 0.5, 24 ± 6 and 96–120 h post-administration. Whole body counting is performed using an uncollimated gamma camera detector.

Dosimetry procedures may be rescheduled up to 1 week when (1) patients fail to comply with the low iodine diet, (2) iodine is accidently administered or ingested (except for iodinated contrast), (3) delivery of I-124 is hampered or (4) PET/CT acquisition and assessment of blood counts and whole body retention is not possible due to failure or maintenance of the hardware and corresponding software. 

#### 2.8.4. I-131 Therapy

The redifferentiation effect will be evaluated by quantification of RAI uptake and assessment of corresponding expected dose to lesion and organs at risk per administered GBq I-131 [Gy/GBq]. Patients are eligible for I-131 therapy with MTA (maximized at 7.4 GBq) if:A clinically meaningful lesion dose is expected: ≥20 Gy in at least one lesion;Acceptable toxicity is expected: blood dose ≤2 Gy and 48 h post-administration body retention ≤3.0 GBq–4.4 GBq (with and without lung metastases).

If considered effective and safe, lenvatinib will be discontinued and standard-of-care I-131 therapy will be given after rhTSH stimulation. Patient preparations are identical to the preparations performed prior to the I-124 dosimetry procedures, including rhTSH administration and dietary restrictions. After ingestion of I-131, patients stay within the hospital considering radiation protection measures and are monitored during their stay until dose-rates are below the local limit for discharge. 

#### 2.8.5. Post-Therapeutic Dosimetry

Post-therapeutic dosimetry is performed for dose verification. Serial SPECT/CT scanning is performed according to Wadsley et al. [28] to determine actual absorbed lesion doses after I-131 therapy. The approach defined by the European Association of Nuclear Medicine (EANM) is used to verify absorbed doses to the blood and whole body retention. [29] Patients undergo SPECT/CT at 24 ± 6, 48 ± 6, 72 ± 6 and 144 ± 24 h post-administration. One heparinized blood sample of 2 mL is withdrawn from the patient and whole body counts are measured at 2–4, 6 ± 1, 24 ± 6, 96 ± 12 and 144 ± 24 h post-administration. 

Quantification of RAI uptake and assessment of corresponding expected absorbed lesion dose per administered GBq I-131 [Gy/GBq] will be determined by dedicated dosimetry software. The absorbed dose to the blood and whole body retention is determined using the approach defined by the EANM [29].

#### 2.8.6. Quality of Life Assessment

QoL is assessed according to 4 different questionnaires:THYCA-QoL: disease-specific health-related QoL questionnaire for thyroid cancer survivors [30];RAND-36 V2.0 (SF36-II): a validated translation in Dutch of the Short Form health survey (SF36-II) comprising 36 questions yielding a 9-scale profile of function health, wellbeing and psychometrically based physical and mental health [31];EQ-5D-5L: EuroQol 5-dimensional 5-level questionnaire: indexing health status including a single visual analogue scale (VAS) [32];DT&PL: an 11-point VAS including a 35-item problem list. We will only use the VAS, since the 35-item problem list mostly overlaps with the content of the other questionnaires [33].

Patients are asked to fill in the surveys at baseline and every time an F-18 FDG PET/CT is performed (at 6, 12, 24 and 36 weeks). In cohort 2B, QoL will also be assessed at 6 weeks. For all patients eligible for I-131 therapy, QoL will be assessed at the last SPECT/CT (Table 1).

#### 2.8.7. Follow-Up

Treatment response will be assessed using standard-of-care (unstimulated) F-18 FDG PET/CT and thyroglobulin levels at 12, 24, 36 and 48 weeks after initiation of lenvatinib treatment as part of the routine evaluation every 12 weeks. This PET/CT replaces the routinely performed contrast-enhanced whole-body CT scan as iodinated (intravenous) contrast media interfere with iodine (I-124, I-131) uptake in thyroid (cancer) tissues. Scans are performed according to EANM guidelines for tumour PET imaging 2.0 [34] and evaluated following PET Response Criteria in Solid Tumors (PERCIST 1.0) [35].

For this study, an additional F-18 FDG PET/CT will be performed at 6 weeks to evaluate the effect of lenvatinib treatment. In all patients treated with I-131, an evaluation will be performed 6 months after I-131 therapy. For cohort 1 this coincides with the standard-of-care follow-up 36 weeks after start of lenvatinib. In cohort 2 this could mean that if the evaluation of cohort 1 confirms that 6 weeks of lenvatinib treatment is the optimal duration, an additional F-18 FDG/PET will be performed at 30 weeks.

### 2.9. Data Analysis and Statistics

The variables obtained before lenvatinib treatment will be compared with those during or after treatment using paired non-parametric tests corresponding to the type of outcome measures. The fraction of patients eligible for I-131 therapy and numeric dosimetry results will be compared using McNemar’s test and the Wilcoxon-signed rank test, respectively. Multiple-comparison correction will be applied when necessary. We will use descriptive statistics to describe metabolic and biochemical treatment responses, incidence and severity of toxicities and quality of life.

An interim analysis is planned when all patients of cohort 1 have finalized their last I-124 PET/CT and a decision is made whether to pursue with I-131 therapy. The interim analysis will be conducted by an independent committee. Based on the primary endpoint of the study (the fraction of patients eligible for I-131 therapy) and I-124 dosimetry results pre- and post-lenvatinib treatment, the optimal treatment duration is determined (either 6 or 12 weeks). Furthermore, incidence and severity of toxicities are evaluated to assess cumulative toxicity of lenvatinib and subsequent I-131 therapy.

## 3. Discussions

In this trial, we evaluate the redifferentiation ability of lenvatinib in RAI-refractory DTC to restore impaired iodine uptake and retention by assessing functional RAI uptake pre- and post-lenvatinib treatment. Following I-124 dosimetry procedures, absorbed doses to lesions and organs at risk will be calculated to determine whether I-131 is deemed effective and safe. This proof-of-concept study poses a potential new treatment approach for RAI-refractory DTC patients by shortening the duration of lenvatinib treatment in combination with subsequent I-131 therapy, thereby potentially reducing severe toxicity and improving quality of life. 

Developments in the identification of RAI-refractory DTC by means of evolving molecular imaging and pathology is pivotal for optimizing current patient management. Assessment of the molecular landscape and altered signaling pathways can help in selecting treatment targets and may lead to a better selection of patients in which a redifferentiation effect is likely to be seen [9,36,37].

Quantification of RAI uptake by means of dosimetry poses the possibility to identify patients who redifferentiate after TKI treatment. Moreover, I-131 therapy can be initiated following a theranostic concept, thereby taking into account the RAI retention of lesions and organs at risk in time [38]. Leading guidelines state that treatments involving ionizing radiation should be individually planned and their delivery verified [39]. While it is standard-of-care in radiotherapy, personalized dosimetry has not yet been widely implemented in nuclear medicine. Estimating absorbed doses to lesions and organs at risk is a time- and resource-consuming process that is prone to errors. As it comprises multiple PET/CT or SPECT/CT measurements, it poses logistical challenges in clinical practice and increases patient and radiation burden. Besides that, there is currently no thorough evidence that dosimetry-based I-131 therapy is more effective than the standard practice approach in which empirical I-131 activities are given to the patient [40]. However, dosimetry is thought to be helpful in the management of more complex patients having a young age, advanced disease, extensive pulmonary metastases, renal insufficiency or failure, suspected recurrence, potential metastatic disease or a combination of these factors [41,42,43]. 

Rigorous and reproducible dosimetry methodologies are needed to facilitate comparison of dosimetry results among studies. To meet this need, we will adhere to the simplified protocols developed by Jentzen et al. for I-124 dosimetry [26,27], guidelines provided by the EANM for intra-therapeutic toxicity assessment [29], and follow initiatives of standardizing quantitative SPECT for dosimetry studies of I-131 treatment [44,45]. 

In the follow-up, we choose to replace the routine 12-weekly evaluation of lenvatinib treatment with contrast-enhanced CT by F-18 FDG-PET/CT. The use of contrast-enhanced CT would preclude diagnostic iodine-scintigraphy (including I-124 PET) and I-131 therapy as iodinated contrast interferes with RAI uptake. This would result in diminished RAI uptake measurements that is undesirable in this trial. Valerio et al. showed that F-18 FDG PET/CT correlates with the response of RAI-refractory DTC to lenvatinib and patient survival [46]. Objective responses are thereby comparable between the two modalities in the long-term and F-18 FDG PET/CT seems even able to predict the long-term response early [46]. Therefore, we see F-18 FDG PET/CT as at least an equivalent alternative to contrast-enhanced CT. 

This trial is aiming at treating patients with short-term lenvatinib treatment with potential subsequent I-131 therapy after identification of a redifferentiation effect. Toxicities of this treatment combination of therapy are not thoroughly investigated, although no cumulative toxicity has been described in comparable trials sequentially combining both selective and multitarget TKIs and I-131 therapy [16,17,18,19,20,21,22,23]. We expect that this treatment combination potentially will avoid severe toxicities that are observed in long-term lenvatinib treatment. The effectiveness of short-term lenvatinib treatment with subsequent I-131 therapy in comparison with long-term lenvatinib treatment is, however, yet to be determined. If our hypothesis that lenvatinib is able to redifferentiate RAI-refractory DTC to renew or increase RAI uptake can be confirmed, the currently established management of RAI-refractory DTC needs to be reconsidered. A combination of short-term lenvatinib treatment with subsequent I-131 therapy may then be preferred as this approach has great potential to reduce toxicity and increase quality of life in comparison to standard-of-care long-term lenvatinib treatment.

## Figures and Tables

**Figure 1 diagnostics-12-03154-f001:**
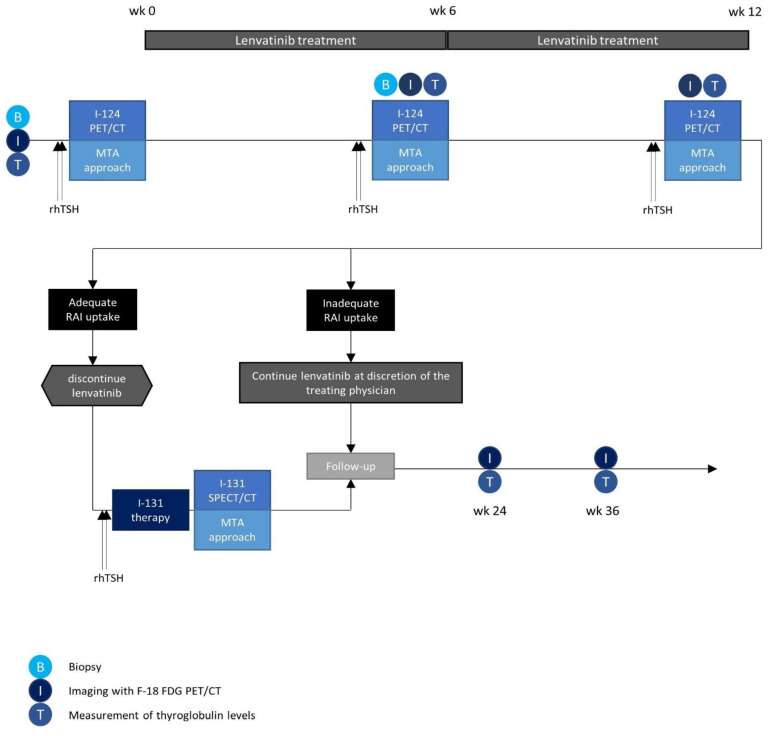
Flowchart of the study for patients in cohort 1, including lenvatinib treatment, dosimetry procedures, I-131 therapy and response evaluation with F-18 FDG PET/CT and serum thyroglobulin levels.

**Table 1 diagnostics-12-03154-t001:** Overview of all (standard and study) procedures in the RESET trial for cohort 1 and cohort 2 in case of 12 and 6-weeks lenvatinib treatment.

12-Wks Lenvatinib		0	1–5	6	7	8–11	12	13	Follow-Up
	Week								(Wk 24, 36)
Medical history		X							
Physical examination		X							
Serum pregnancy test (females)		X		X ^iii^			X	X ^i,ii^	
Hematology and blood chemistry **^a^**		X							
Biopsy		X ^i^		X					
Lenvatinib treatment			X ^i^	X ^i^	X ^i^	X ^i^	X ^i^		
rhTSH-stimulated I-124 dosimetry ^b^		X		X ^iii^			X		
rhTSH-stimulated I-131 therapy ^c^								X ^i,ii^	
Post-therapeutic I-131 dosimetry ^d^								X ^ii^	
Thyroid Hormone Levels		X		X ^iii^			X	X ^i,ii^	
Unstimulated Tg levels		X ^i^		X ^i^			X ^i^		X ^i^
F-18 FDG PET/CT ^e^		X ^i^		X ^iii^			X ^i^		X ^i^
QoL assessment		X		X			X	X ^ii^	X
**6-Wks Lenvatinib**									**Follow-up**
	**Week**	**0**	**1–5**	**6**	**7**	**8–11**	**12**	**13**	**(24, 30, 36)**
History		X							
Physical examination		X							
Serum pregnancy test (females)		X		X	X ^i,ii^				
Hematology and blood chemistry **^a^**		X							
Biopsy		X ^i^		X					
Lenvatinib treatment			X ^i^	X ^i^					
rhTSH-stimulated I-124 dosimetry ^b^		X		X					
rhTSH-stimulated I-131 therapy ^c^					X ^i,ii^				
Post-therapeutic I-131 dosimetry ^d^					X ^ii^				
Thyroid Hormone Levels		X		X	X ^i,ii^				
Unstimulated Tg levels		X ^i^		X ^i^			X ^i^		X ^i^
F-18 FDG PET/CT ^e^		X ^i^		X			X ^i^		X ^i^
QoL assessment		X		X	X ^ii^		X		X

^i^: standard procedures, ^ii^: only provided to patients eligible for I-131 therapy; ^iii^: only provided to patients in cohort 1.^a^: Hematology: Hb, platelet count, absolute neutrophil count (ANC), white blood cell diff, hematocrit, PT/INR, APTT. Chemistry: LDH, phosphorus, sodium, potassium, magnesium, chloride, calcium, creatinine, albumin, total protein, SGOT (AST), SGPT (ALT), bilirubin (indirect + direct), GGT, alkaline phosphatase, glucose, amylase, lipase, TSH, fT4, thyroglobulin. ^b^: prior to I-124 administration of 37 (±10%) MBq, 0.9 mg Thyrotropin alfa injections are given on 2 consecutive days. At 24 ± 6 and 96–120 h post-administration of I-124, PET/CT imaging is performed. Blood draws and whole body counting are performed at 2–4, 24 ± 6 and 96–120 h post-administration. ^c^: prior to I-131 administration, 0.9 mg Thyrotropin alfa injections are given on 2 consecutive days. The MTA (≤7.4 GBq) I-131 is administered to the patient after which a stay in the hospital is required for an expected 4–5 days. ^d^: SPECT/CT imaging is performed at 24 ± 6, 48 ± 6, 72 ± 6 and 144 ± 24 h post-administration. Blood draws and whole-body counting will be performed at 2–4, 6 ± 1, 24 ± 6, 96 ± 12 and 144 ± 24 h post-administration. ^e^: At 12, 24 and 36 weeks after initiation of lenvatinib, F-18 FDG PET/CT will be performed. In all patients treated with I-131, an evaluation will also be performed 6 months after I-131 therapy.

**Table 2 diagnostics-12-03154-t002:** Inclusion- and exclusion criteria of the RESET trial.

Inclusion Criteria	
General	Age ≥18 years ECOG performance status ≤2 Life expectancy ≥3 months Agreement to use a highly effective method of contraception during the study
Tumor characteristics	Histologically or cytologically confirmed DTC Progressive (biochemical or anatomic) disease for which lenvatinib is indicated RAI-refractory disease on structural imaging, defined as: - Metastatic lesions that are not RAI-avid on a diagnostic or intra-therapeutic RAI scan performed prior to enrolment in the current study - RAI-avid metastatic lesions which remained stable in size or progressed according to RECIST 1.1 criteria despite RAI treatment. Absence of response is observed during 6–9 months after high dose I-131 therapy. Measurable disease on ceCT prior to inclusion according to RECIST 1.1: - ≥1.0 cm in the longest diameter for a non-lymph node - ≥1.5 cm in the short axis for a lymph node
Blood levels	Creatinine levels indicating renal clearance ≥50 mL/min INR ≤ 1.5 Absolute neutrophil count ≥1.5 × 10^9^/L Hemoglobin ≥9 g/dL (5.6 mmol/L) Platelets ≥100 × 10^9^/L Albumin ≥25 g/L Total bilirubin <1.5× ULN ASAT ≤3× ULN (≤5× ULN in case of liver metastases) ALAT ≤3× ULN (≤5× ULN in case of liver metastases)
**Exclusion Criteria**	
General	Known hypersensitivity or idiosyncrasy to lenvatinib and/or to thyrotropin alfa Received iodinated intravenous contrast <6–8 weeks prior to enrollment Inability to follow a low iodine diet Pregnant, lactating or breast-feeding women Inability to give informed consent
Concomitant diseases	Other malignancies <3 years prior to enrollment, except completely resected non-melanoma skin cancer or indolent secondary malignancies Symptomatic or untreated leptomeningeal or brain metastases or spinal cord compression Gastrointestinal abnormalities that may alter absorption Evidence of cardiovascular risk including clinically relevant arrhythmias, acute coronary syndromes, severe/unstable angina or symptomatic congestive heart failure Concurrent uncontrolled medical condition
Treatments	I-131 therapy <6 months prior to enrollment External beam radiation therapy <4 weeks prior to start lenvatinib Treatment with investigational drugs <4 weeks prior to start lenvatinib Requiring medication with high content in iodide (e.g., amiodarone)

## Data Availability

The results of the RESET trial will be disseminated after completion in peer-reviewed journals and presented at conferences. Data generated and analyzed within the RESET trial shall only be shared upon request with researchers who provide a methodologically sound research proposal, at the discretion of the principal investigators. Only de-identified participant data from the final research dataset used in the published manuscript can be shared.

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
