# Peer review of "Reinducing Radioiodine-Sensitivity in Radioiodine-Refractory Thyroid Cancer Using Lenvatinib (RESET): Study Protocol for a Single-Center, Open Label Phase II Trial"

_diagnostics, 2022, doi:10.3390/diagnostics12123154_

Round 1
Reviewer 1 Report
The article is well written, the introduction provides sufficient background and introduces the discussed issue. The study protocol is described properly, only the dosimetry procedures could be described in more detail.
However, I am not convinced for the publication only of the protocol of the study, without any results. In my opinion this is just very good, first part of the manuscript and it should be published together with results, when they appear. Otherwise, later the reader will have to look for a description of the protocol, because it will be difficult to describe it in a shorter form again. Such a division is artificial.
I noticed the manuscript is of protocol type. Therefore, I made the final decision to: accepted with minor revision.
Author Response
Dear reviewer,
First of all, we would like to thank you for reviewing the manuscript comprising the clinical study protocol of the RESET trial. This cover letter encloses a point-by-point response to the comments and suggestions.
Point 1 Dosimetry procedures could be described in more detail
Comment: we kept the dosimetry procedures short and practical as they are covered elaborately in the articles we refer to. We also did not want the reader to get lost in somewhat more complex formulas and technical details that accompany I-124 and I-131 dosimetry. We think it is the most important that the reader is aware this trial is about evaluating the redifferentiation effect of lenvatinib in radioiodine-refractory cancer and that iodine uptake is quantified following known procedures. If the reader is more interested in the details of these procedures, they can read the articles that we refer to in both the study procedures section and the discussion.
Point 2 The manuscript should be published together with the results when they become available
Comment: as you kindly mentioned in your review, this is indeed a protocol article. Publishing this protocol gives us the opportunity to announce the research to the community even further to increase our patient recruitment. As there is a limited amount of patients with radioiodine-refractory thyroid cancer in the Netherlands, the end results of this trial will not be available until at least 2024.
Thank you for your critical view on this manuscript. We appreciate your comments and feedback.
Yours sincerely,
The RESET study team
Reviewer 2 Report
The paper deals with a very topical clinical question. The definition of refractory iodine is controversial, not all the established criteria are accepted since the limit of activity to be administered is arbitrary and the stability of the disease after 131-I administration cannot be considered refractory.The use of FDG PET for diagnosis and follow-up may not be adequate since not all lesions need to pick up FDG. For follow-up, iodine uptaking lesions should be correlated with FDG uptaking lesions, as there may be missmatch lesions at diagnosis.
In addition to assessing cell redifferentiation with 131I uptake,morphologycal response should be assessed, which is what is clinically relevant, and nonuptake of FDG is not included in the response criteria. This is even more relevant if lesions that uptake 131I and remain stable have been considered as refractory.
The paper presents a theoretical approach but does NOT present results.
Author Response
Dear reviewer,
First of all, we would like to thank you for reviewing the manuscript comprising the clinical study protocol of the RESET trial. This cover letter encloses a point-by-point response to the comments and suggestions.
Point 1 The definition of radioiodine-refractoriness is controversial and not all established criteria are accepted since the limit of activity to be administered is arbitrary and the stability of disease after I-131 administration cannot be considered refractory.
Comment: there are two important inclusion criteria within the trial: (1) the patient is considered radioiodine-refractory by the medical board and (2) there is an indication for lenvatinib. The medical board holds on to the established (yet controversial) criteria and the patient is considered non-responsive to prior I-131 therapy when he enters the trial (based on clinical parameters including Tg-measurements and anatomical imaging). As for the second one, patients with locally advanced or metastatic disease have to be progressive before they can start with lenvatinib and local therapy (e.g. palliative resection, EBRT), should have been considered as alternative non-systemic approach. There are no patients in this trial who are considered ‘stable disease’ (as defined by RECIST criteria).
Point 2 The use of FDG PET/CT for diagnosis and follow-up may not be adequate since not all lesions need to pick up FDG. Morphological response is what is clinically relevant and non-uptake of FDG is not included in the response criteria.
Comment: the current guidelines state that contrast enhanced CT in combination with the RECIST criteria have to be used for (morphological) response evaluation in patients with radioiodine-refractory thyroid cancer who are receiving lenvatinib treatment. As contrast contains iodine and therefore interferes with the radioactive iodine administered to the patient for the dosimetry procedures and I-131 therapy (if possible), this evaluation method is not absolutely contraindicated for this trial. As mentioned in the discussion, Valerio et al. [1] show that FDG PET/CT is capable of predicting early and long-term response to lenvatinib treatment and correlates with overall survival. Moreover, FDG uptake is expected in most dedifferentiated lesions that show no I-131 uptake, which is in line with our own observations.[2-4] Rendl et al. [5] also show that FDG PET/CT is feasible for assessing treatment response during lenvatinib treatment. Nonetheless, lesions that are negative for both FDG and radioactive iodine will be identified at baseline and every follow-up scan. As a non-contrast enhanced CT is performed with each PET scan, both FDG and anatomical response will be assessed for each patient. Since there is no iodinated contrast that will be given to the patient, the latter will somewhat be more difficult in case of liver and lymph node metastases in comparison to the current standard. Finally, after the last I-124 dosimetry procedure or I-131 therapy, further follow-up will be performed using contrast-enhanced CT. FDG-PET/CT will only replace contrast-enhanced CT (RECIST) during the first 13 weeks of the study.
[1] Valerio L, Guidoccio F, Giani C, Tardelli E, Puccini G, Puleo L, Minaldi E, Boni G, Elisei R, Volterrani D. [18F]-FDG-PET/CT Correlates With the Response of Radiorefractory Thyroid Cancer to Lenvatinib and Patient Survival. J Clin Endocrinol Metab. 2021 Jul 13;106(8):2355-2366. doi: 10.1210/clinem/dgab278. PMID: 33901285.
[2] Feine U, Lietzenmayer R, Hanke JP, Held J, Wöhrle H, Müller-Schauenburg W. Fluorine-18-FDG and iodine-131-iodide uptake in thyroid cancer. J Nucl Med. 1996 Sep;37(9):1468-72. PMID: 8790195.
[3] Lodi Rizzini E, Repaci A, Tabacchi E, Zanoni L, Vicennati V, Cavicchi O, Pagotto U, Morganti AG, Fanti S, Monari F. Impact of 18F-FDG PET/CT on Clinical Management of Suspected Radio-Iodine Refractory Differentiated Thyroid Cancer (RAI-R-DTC). Diagnostics (Basel). 2021 Aug 7;11(8):1430. doi: 10.3390/diagnostics11081430. PMID: 34441364; PMCID: PMC8391566.
[4] Ha LN, Iravani A, Nhung NT, Hanh NTM, Hutomo F, Son MH. Relationship between clinicopathologic factors and FDG avidity in radioiodine-negative recurrent or metastatic differentiated thyroid carcinoma. Cancer Imaging. 2021 Jan 7;21(1):8. doi: 10.1186/s40644-020-00378-z. PMID: 33413689; PMCID: PMC7792294.
[5] Rendl G, Schweighofer-Zwink G, Sorko S, Gallowitsch HJ, Hitzl W, Reisinger D, Pirich C. Assessment of Treatment Response to Lenvatinib in Thyroid Cancer Monitored by F-18 FDG PET/CT Using PERCIST 1.0, Modified PERCIST and EORTC Criteria-Which One Is Most Suitable? Cancers (Basel). 2022 Apr 7;14(8):1868. doi: 10.3390/cancers14081868. PMID: 35454777; PMCID: PMC9029268.
Thank you for your critical view on this manuscript. We appreciate your comments and feedback.
Yours sincerely,
The RESET study team
Round 2
Reviewer 2 Report
Accept in present form